# Down-Regulation of HLA-C Expression on Melanocytes May Contribute to the Therapeutic Efficacy of UVB Phototherapy in Psoriasis

**DOI:** 10.3390/ijms26072858

**Published:** 2025-03-21

**Authors:** Yukiyasu Arakawa, Akiko Arakawa, Seçil Vural, Mengwen He, Sigrid Vollmer, Jörg C. Prinz

**Affiliations:** Department of Dermatology and Allergy, University Hospital, Ludwig-Maximilian-University Munich, D-80337 Munich, Germany; aara@koto.kpu-m.ac.jp (Y.A.); akikoxxx999@gmail.com (A.A.); sevural@ku.edu.tr (S.V.); hemwtj@126.com (M.H.); sigrid.vollmer@med.uni-muenchen.de (S.V.)

**Keywords:** psoriasis, UVB phototherapy, HLA-class I expression, melanocyte immunogenicity, HLA-C*06:02

## Abstract

UVB phototherapy effectively treats psoriasis. Although it suppresses both innate and adaptive immunity, it remains unclear why UVB irradiation is primarily effective for T-cell-mediated but not inflammatory skin diseases of other etiologies. Using a Vα3S1/Vβ13S1 T-cell receptor (TCR) from a lesional psoriatic CD8^+^ T-cell clone, we recently demonstrated that in psoriasis, the major psoriasis risk allele HLA-C*06:02 mediates an autoimmune response of CD8^+^ T-cells against melanocytes by presenting a melanocyte autoantigen. We now investigate the effect of UVB irradiation on melanocyte immunogenicity using the psoriatic Vα3S1/Vβ13S1 TCR in a reporter assay. The immunogenicity of melanocytes for the Vα3S1/Vβ13S1 TCR depended on the up-regulation of HLA-C expression by IFN-γ. UVB irradiation reduced the stimulatory capacity of IFN-γ-conditioned melanocytes for the Vα3S1/Vβ13S1 TCR by suppressing key IFN-γ-induced MHC-class I transcriptional regulators (STAT1, IRF1, NLRC5), the HLA-C-specific transcription factor Oct1, and by inducing miR-148a, which specifically inhibits HLA-C expression. This resulted in the suppression of the IFN-γ-induced expression of HLA-class I molecules and, in particular, an almost complete loss of HLA-C expression. We conclude that suppression of the inflammatory increase in HLA-class I expression and antigen-presentation may contribute to the efficacy of UVB phototherapy in T-cell-mediated skin diseases. The pronounced downregulation of HLA-C on melanocytes could render psoriasis, as HLA-C-associated disease, particularly susceptible to this effect.

## 1. Introduction

Phototherapy with UVB radiation is a widely used treatment modality for various immune-mediated skin diseases. It refers to the therapeutic exposure of the skin to either broadband (290 to 320 nm) or narrowband (311 to 313 nm) UVB radiation. The immunosuppressive effects of UVB irradiation are achieved by impairing both innate and adaptive immunity [1] through multiple molecular and cellular changes that occur when chromophores in the skin absorb the energy of the radiation [2]. Despite its multiple effects, however, UVB therapy is mainly effective in T-cell-mediated skin diseases such as psoriasis, the most frequent indication for this treatment worldwide [3,4,5]. (Auto)antibody-mediated skin diseases or inflammatory skin diseases of other etiologies hardly respond to it. Accordingly, T-cell-mediated immune responses should exhibit a distinctive feature that renders them particularly susceptible to the therapeutic mechanisms of UVB irradiation relative to other inflammatory skin conditions.

Psoriasis is a T-cell-mediated autoimmune skin disease. The main psoriasis HLA risk allele is HLA-C*06:02 [6,7,8]. Less frequently associated HLA-class I-alleles which also confer susceptibility to psoriasis are HLA-C*07:01, HLA-C*07:02, HLA-C*12:02, HLA-C*07:04, HLA-B*27, and HLA-B*57 [9,10,11,12,13]. Formation of psoriasis plaques results from the epidermal recruitment, activation, and clonal expansion of CD8^+^ T-cells [14,15,16]. Recent evidence has clearly demonstrated that these epidermal CD8^+^ T-cells are the actual producers of the psoriasis signature cytokines IL-17, IL-22, and TNF-α that convey the skin changes characteristic of psoriasis [17,18,19,20]. Unlike the cytotoxic CD8^+^ T-cells that eradicate melanocytes in vitiligo, the T_c_17 phenotype of the lesional psoriatic CD8^+^ T-cells promotes local inflammation through the production of IL-17 [21]. Further, T-cell-derived IL-17 and TNF-α synergize in stimulating the proliferation of melanocytes while simultaneously suppressing pigmentation-related genes and lowering cellular tyrosinase levels. This increases the lesional density of melanocytes in psoriasis lesions but reduces their cellular melanin content [22,23]. The increased epidermal melanocyte density, together with impaired melonogenesis, is held responsible for the post-inflammatory hyper- and hypopigmentation often observed in resolved psoriasis lesions [24].

In studies on psoriasis, UVB irradiation suppressed the epidermal expression of IL-17, IL-22, and IFN-γ and rapidly depleted the intraepidermal T-cells. These effects correlated strongly with the resolution of psoriasis lesions and thus provided evidence that suppressing the activation of epidermal T-cells is part of the mechanism of action of UVB phototherapy [25,26,27,28,29,30]. Transcriptome profiling of full-thickness psoriasis skin biopsies revealed significant changes in differentially expressed genes following UVB irradiation that involved suppression of the NF-κB pathway and up-regulation of the sirtuin signalling cascade, with the enrichment of keratinization-related genes reflecting normalized skin physiology [31,32].

Using a paradigmatic Vα3S1/Vβ13S1 TCR from an epidermal CD8^+^ T-cell clone isolated from skin lesions of an HLA-C*06:02^+^ psoriasis patient [15] and expressed together with CD8 in the 58α^−^/β^−^ reporter mouse T hybridoma cell line [33], we previously demonstrated that HLA-C*06:02 orchestrates an autoimmune response of epidermal CD8^+^ T-cells against melanocytes. This Vα3S1/Vβ13S1 TCR specifically recognizes a self-peptide derived from ADAMTS-like protein 5 (ADAMTSL5) as a melanocyte autoantigen presented by HLA-C*06:02, engaging in a unique charge network with exposed arginine residues from both the self-peptide and the HLA-C*06:02 α1-helix [34,35]. Significantly increased blood levels of ADAMTSL5-specific CD8^+^ T-cells and ADAMTSL5 autoantibodies in psoriasis patients substantiate the role of ADAMTSL5 as a psoriatic autoantigen [36,37]. We further demonstrated that the autoantigenic ADAMTSL5 epitope is generated through NH_2_-terminal trimming of precursor peptides by the endoplasmic reticulum aminopeptidase 1 (ERAP1) [38], clarifying the functional consequences of the gene–gene interaction between HLA-C*06:02 and ERAP1 variants in psoriasis susceptibility [39]. The immunogenicity of melanocytes depends on IFN-γ, which is abundantly expressed in psoriasis lesions and upregulates HLA-C and ERAP1 that are both highly expressed by melanocytes in the basal epidermal layers of psoriasis lesions [34,38,40,41]. Hence, the generation of the psoriatic autoantigen by ERAP1 and its presentation by HLA-C*06:02 on melanocytes are essential events in the induction of the autoimmune response by CD8^+^ T-cells in psoriasis [42].

To investigate the mechanisms underlying the efficacy of phototherapy in psoriasis, we have now examined the impact of UVB irradiation on the ability of melanocytes to stimulate the HLA-C*06:02-restricted Vα3S1/Vβ13S1 TCR in an in vitro assay that models the autoimmune response in psoriasis. We find that UVB irradiation significantly reduces the IFN-γ-dependent immunogenicity of melanocytes for the Vα3S1/Vβ13S1 TCR by downregulating HLA-C expression. This reduction was mediated by suppression of the IFN-γ-induced STAT1-IRF1-NLRC5 signalling pathway of HLA-class I antigen presentation and by interference with HLA-C-specific transcriptional and post-translational regulators. The mode of action of UVB phototherapy in psoriasis could therefore involve depriving the pathogenic CD8^+^ T-cells of the stimulatory signal in the epidermis by suppressing the HLA-class I antigen processing and presentation pathway. This could render T-cell-mediated skin diseases particularly sensitive to the effects of UVB irradiation.

## 2. Results

### 2.1. UVB Irradiation of Melanocytes Suppresses Their Ability to Stimulate the Vα3S1/Vβ13S1 TCR

We investigated the impact of UVB irradiation on the ability of HLA-C*06:02-positive primary human melanocytes (PHMs) to stimulate the autoreactive ADAMTSL5-specific psoriatic Vα3S1/Vβ13S1 TCR in a TCR activation assay. In this assay, melanocytes are co-cultered with the Vα3S1/Vβ13S1 TCR hybridoma, whereby TCR ligation by the HLA-C*06:02/ADAMTSL5 complex induces the formation of superfolder green fluorescent protein (sGFP) that can be measured using flow cytometry [34,43]. The percentage of sGFP^+^ hybridoma cells directly reflects the level of TCR stimulation [38].

The immunogenicity of melanocytes for the HLA-C*06:02-restricted Vα3S1/Vβ13S1 TCR is dependent on IFN-γ [34]. It enhances the generation and presentation of the ADAMTSL5 peptide by up-regulating the otherwise low HLA-C surface expression [34] and increasing the expression of ERAP1 [44,45]. Therefore, in our assay, PHMs were grown in 48-well microtiter plates with or without IFN-γ to near confluence and then irradiated with 50 or 100 mJ/cm^2^ UVB. Subsequently, Vα3S1/Vβ13S1 TCR hybridoma cells were added, and after 24 h of co-culture, TCR stimulation was determined by measuring the sGFP-positive hybridoma cells using flow cytometry. This experimental system allows for an analysis of the effect of UVB specifically on melanocytes as there are no other cellular components that could influence the immunogenicity of melanocytes and TCR stimulation by chemokines or cytokines such as TGF-β or IL-4.

Consistent with our previous findings [34,38], IFN-γ-conditioned PHMs strongly stimulated the Vα3S1/Vβ13S1 TCR (Figure 1A). The applied UVB doses of 50 mJ/cm^2^ or 100 mJ/cm^2^ significantly reduced the IFN-γ-dependent TCR stimulation in a dose-dependent manner (*p* < 0.05). This decrease was not due to UVB-induced apoptosis, as UVB dose escalation up to 200 mJ/cm^2^ hardly affected melanocyte viability (Appendix A).

### 2.2. The Effect of UVB on Melanocyte Immunogenicity Can Be Reproduced Using HLA-C*06:02-Positive Melanoma Cell Lines

HLA-C*06:02-positive melanoma cell lines can substitute for normal human melanocytes in the analysis of the melanocyte-specific reactivity of the autoreactive Vα3S1/Vβ13S1 TCR [34]. We employed three different HLA-C06:02^+^ melanoma cell lines, namely, 1205LU, WM793, and WM278, to validate the effects of UVB on melanocyte immunogenicity. All three cell lines stimulated the Vα3S1/Vβ13S1 TCR when preconditioned with IFN-γ. Similar to PHMs, irradiation of IFN-γ-conditioned 1205Lu (Appendix A), WM793 (Appendix A), or WM278 cells (Appendix A) with UVB 50 mJ/cm^2^ or 100 mJ/cm^2^ significantly reduced the activation of the Vα3S1/Vβ13S1 TCR. Notably, this effect was exclusive to UVB radiation, as irradiation of the melanoma cell lines WM278 und 1205LU with UVA (340–400 nm, 3 or 7.5 J/cm^2^) did not affect Vα3S1/Vβ13S1 TCR stimulation (WM278, 1205Lu) and HLA-C expression (1205LU) by the melanoma cell lines tested (Appendix A). Quantitative PCR analysis revealed that neither IFN-γ nor UVB radiation had a significant impact on mRNA transcript levels of the melanocyte autoantigen, ADAMTSL5, in PHMs (Figure 1B) or melanoma cell lines (Appendix A). Thus, UVB radiation decreased the ability of melanocytes to activate the Vα3S1/Vβ13S1 TCR without affecting autoantigen expression.

### 2.3. UVB Suppresses IFN-γ-Induced HLA-Class I and HLA-C Expression on Melanocytes

We next investigated whether UVB irradiation affected HLA-class I surface expression. PHMs and melanoma cell lines were cultured with or without IFN-γ, exposed to 50 or 100 mJ/cm^2^ UVB and the expression of HLA-class I molecules (HLA-ABC) and HLA-C analyzed by flow cytometry and fluorescence microscopy. Compared to the overall HLA-class I expression (HLA-ABC), the baseline level of HLA-C on the cell surface was low, as described [46]. IFN-γ treatment induced an upregulation of the HLA-class I molecules on PHMs (Figure 1C) and melanoma cell lines (Appendix A), which was particularly prominent for HLA-C. UVB irradiation of PHMs did not significantly alter the basal expression of HLA-ABC or HLA-C. For IFN-γ-conditioned PHMs, UVB irradiation reduced the IFN-γ-induced up-regulation of HLA-class I molecules in a dose-dependent manner (Figure 1C–E). This decline was particularly evident for HLA-C expression, which returned to basal expression levels. Similar results were observed for the melanoma cell lines (Appendix A). This suggests that UVB primarily affected the IFN-γ-induced rather than the constitutive expression of the HLA-class I molecules. UVA irradiation had no significant effect on IFN-γ-induced HLA-C expression, while it even appeared to increase the IFN-γ-induced total HLA-class I expression (Appendix A).

### 2.4. UVB Down-Regulates IFN-γ-Induced Translational and Posttranscriptional Regulators of HLA-Class I and HLA-C Expression

To investigate the mechanisms underlying the down-regulation of HLA-C cell surface expression, we measured HLA-C mRNA levels using quantitative PCR. IFN-γ significantly increased HLA-C mRNA levels in primary human melanocytes (PHMs) and melanoma cell lines. These levels decreased significantly already 3 h after UVB irradiation (Figure 1F, Appendix A), indicating that UVB inhibits HLA-C expression at the transcriptional level.

IFN-γ is the main initiator of inducible HLA-class I expression. Key transcriptional regulators of the IFN-γ signaling pathway are STAT1 (Signal Transducer and Activator of Transcription 1), IRF1 (Interferon Regulatory Factor 1), and NLRC5 (NLR Family CARD Domain Containing 5) [47,48,49,50]. The level of HLA-C expression is then further modulated by the transcription factor Oct1, which may enhance HLA-C expression [51], and by miR-148, which suppresses the expression of HLA-C alleles with an intact binding site in the 3’-untranslated region of HLA-C [52].

An analysis of transcript levels in PHMs by quantitative PCR revealed low basal mRNA expression for STAT1, IRF1, and NLRC5 that was minimally affected by UVB irradiation but strongly increased by IFN-γ (Figure 2A–C). UVB irradiation significantly reduced IFN-γ-induced mRNA levels of all three transcriptional regulators, which for STAT1 and IRF1 was already evident within 3 hrs. The silencing of NLRC5 using siRNA exemplarily demonstrated the consequences of UVB-induced suppression of the HLA-class I transactivators. It significantly decreased the IFN-γ-enhanced expression of HLA-C and immunogenicity of the melanoma cell lines, 1205Lu, WM278, and WM793 for the Vα3S1/Vβ13S1 TCR (Figure 3A–D). The expression of HLA-ABC was only affected by the knockdown of β2-microglobulin (B2M), an essential component for the expression of the heterodimeric HLA-class I proteins, but not NLRC5 (Appendix A).

The transcript level of Oct1 was not affected by IFN-γ, but was significantly reduced by UVB irradiation, regardless of the presence or absence of IFN-γ (Figure 2D). IFN-γ or UVB irradiation alone moderately decreased the expression of miR-148a. However, in the presence of IFN-γ, UVB radiation (50 mJ/cm^2^) was a significant inducer of miR-148a in PHMs (Figure 2E), thus counteracting the IFN-γ-mediated amplification of HLA-C expression.

Regulation of cell surface expression of HLA-C further involves ERAP1 and B2M. ERAP1 and B2M transcription were up-regulated by IFN-γ and down-regulated by UVB irradiation within 3 h, with a rebound-like increase in B2M mRNA observed after 24 h (Figure 2F,G). The instrumental role of B2M in HLA-C expression was confirmed by silencing B2M, which abolished HLA-ABC and HLA-C expression and Vα3S1/Vβ13S1 TCR stimulation (Figure 3A–D, Appendix A). Thus, UVB down-regulates the IFN-γ-induced expression of HLA-class I molecules and in particular of HLA-C through interfering with various transcriptional and posttranslational events.

### 2.5. The Immunogenicity of Melanocytes for the Vα3S1/Vβ13S1 TCR Is Lost with Suppression of HLA-C Expression

To investigate whether UVB-induced suppression of HLA-C expression caused the loss of Vα3S1/Vβ13S1-TCR stimulation by melanocytes, we knocked down HLA-C in 1205Lu, WM278, and WM793 cells with two HLA-C-targeting siRNA oligonucleotides. This largely abolished both HLA-C expression and Vα3S1/Vβ13S1-TCR activation (Figure 4A–C). Thus, suppression of HLA-C is indeed the mechanism by which UVB irradiation reduces the immunogenicity of melanocytes for the Vα3S1/Vβ13S1 TCR.

Finally, in a first attempt, we searched for initial evidence of whether UVB phototherapy can reduce HLA-C expression on melanocytes in psoriasis lesions. In agreement with previous findings [38], immunohistologic analysis of HLA-C expression in untreated lesional psoriatic skin revealed a high expression of HLA-C in the basal epidermal layer, which was particularly pronounced on melanocytes (Figure 4D,E). After phototherapy of a psoriasis patient with narrow-band UVB (41 irradiations, cumulative dose of 44.1 J/cm^2^), the melanocytes in a healed psoriasis plaque had almost completely lost the expression of HLA-C, as evidenced by a strongly reduced mean fluorescence intensity of HLA-C staining in c-kit-positive areas. This tentative observation suggests that suppressing the pathway of HLA-class I expression and antigen presentation is indeed a potential mechanism of action of UVB phototherapy.

## 3. Discussion

Using an experimental model that reproduces the HLA-C*06:02-restricted autoimmune response of CD8^+^ T-cells against melanocytes in psoriasis, we identified a potential mechanism of action of UVB phototherapy. We find that UVB irradiation suppresses the IFN-γ-driven STAT1-IRF1-NLRC5 pathway of HLA-class I expression and antigen presentation and further affects HLA-C-specific transcriptional regulators. This results in a significant reduction in IFN-γ-induced HLA-class I expression, especially lowering HLA-C levels. As a result, melanocytes lose their immunogenicity for the psoriatic autoimmune response. UVB phototherapy of psoriasis can thus deprive the pathogenic CD8^+^ T-cells of stimulatory signals in the epidermis.

HLA-class I molecules present peptide antigens from cytoplasmic proteins to the TCRs of CD8^+^ T-cells and thus direct immune responses against target cells expressing the parental proteins of the antigenic peptides [53]. High HLA-class I expression enhances the strength of CD8^+^ T-cell responses, and overexpression of HLA-class I molecules is associated with both better control of viral infections and the onset of autoimmune diseases, an effect particularly prominent for HLA-C [54,55,56,57]. In psoriasis, HLA-C is highly expressed on melanocytes in the basal epidermal layers of psoriatic plaques [38].

In our study, we observed that UVB irradiation reduced the IFN-γ-induced HLA-class I expression on melanocytes, bringing it to approximately the basal expression levels observed without IFN-γ exposure. This indicates that UVB primarily attenuates the inflammatory, i.e., IFN-γ-induced rather than the constitutive expression of HLA-class I molecules. Given the very low basal expression of HLA-C on melanocytes and its reliance on IFN-γ for upregulation, the reduction in HLA-C expression was particularly significant relative to overall HLA-ABC expression. As a result, melanocytes lost the IFN-γ-dependent capacity to stimulate the Vα3S1/Vβ13S1 TCR.

IFN-γ is the main activator of the inflammatory HLA-class I antigen presentation. Binding of IFN-γ to its receptor on the cell membrane induces phosphorylation of STAT1 by the janus kinases, JAK1 and JAK2. STAT1 is a central transcriptional regulator of psoriatic inflammation and highly expressed in psoriasis lesions [58,59,60]. Activated STAT1 initiates the transcription of IRF1 and NLRC5 in the nucleus. IRF1 binds to the interferon-stimulated response element (ISRE) region in the MHC I promotor on MHC-class I genes [61] and further regulates the constitutive and IFN-γ-induced expression of ERAP1 [62,63]. NLRC5 is the master transcriptional activator of MHC-class I gene expression and also coordinates the expression of B2M [47,64,65]. Up-regulation of NLRC5 strongly enhances MHC-class I-mediated CD8^+^ T-cell responses [61].

Accordingly, IFN-γ strongly upregulated the transcription of STAT1, NLRC5, and IRF1 as well as the expression of HLA-class I by melanocytes. We now find that the decrease in HLA-class I expression by UVB irradiation was due to a suppression of the IFN-γ-induced STAT1 expression and the downstream IRF1-NLRC5 signaling pathway. This corresponds to the decrease in STAT1 expression in psoriasis lesions observed after UVB phototherapy by transcriptome profiling, although this finding could not be attributed to a particular cell type in the whole skin biopsies [31]. Exemplary knockdown of NLRC5 and B2M by siRNA in three different melanoma cell lines confirmed that downregulation of the IFN-γ-induced STAT1 pathway indeed significantly impairs HLA-C expression and Vα3S1/Vβ13S1 TCR stimulation. Suppression of this signaling pathway by UVB should also cause the down-regulation of ERAP1, which is controlled by IRF1 and essential for the generation of the ADAMTSL5 epitope from precursor peptides [38]. UVB thus inhibits the generation and presentation of the autoantigenic ADAMTSL5 peptide at two levels: by suppressing both HLA-C*06:02 and ERAP1 expression. Suppression of NLRC5 by UVB should likewise reduce the overall expression of HLA-class I molecules induced by IFN-α [66], although knockdown of NLRC5 alone did not cause any change.

Beyond its impact on IFN-γ-induced HLA-class I expression, the immunosuppressive activity of UVB also exerted HLA-C-specific effects. UVB repressed Oct1, an IFN-γ-independent transcription factor for HLA-C that modulates HLA-C expression [51] and, similar to STAT1 and IRF1, is a transcriptional regulator of differentially expressed genes in psoriasis lesions [67]. Furthermore, UVB strongly induced the formation of miR-148a in the presence of IFN-γ, thereby counteracting the IFN-γ-induced inflammatory up-regulation of HLA-C. The binding of miR-148a to its target site within the 3’ untranslated region of HLA-C specifically impairs the transcription of HLA-C alleles that bind this microRNA [52]. However, the HLA-C*06:02 allele lacks a binding site for miR-148a. Therefore, the increase in miR-148a expression is not relevant for the HLA-C*06:02-restricted immunity of melanocytes in our assay. Its induction is a potential mechanism contributing to the immunosuppressive effects of UVB for HLA-C alleles with a corresponding binding site.

UVB suppressed the IFN-γ-mediated increase in the expression of HLA-ABC in the PHMs to a lesser extent than in the melanoma cell lines. This could be related to the fact that the expression of HLA class I molecules is less stable in malignant melanomas and in melanoma cell lines [68,69]. Consequently, the HLA-A and HLA-B expressions in the melanoma cell lines might be more susceptible to UVB irradiation than in PHMs. Furthermore, the knockdown of NLRC5 reduced HLA-C expression but hardly affected the expression of HLA-ABC, indicating a differential regulation. In principle, these observations might further support that an HLA-C-restricted immune response in particular is more sensitive to UVB irradiation.

The study has several limitations. The markedly stronger reduction in HLA-C compared to the overall HLA-class I expression by UVB is relevant not least because, in addition to HLA-C*06:02, other HLA-C alleles, namely, HLA-C*07:01, HLA-C*07:02, and HLA-C*07:04, may also confer a risk for psoriasis, possibly because they belong to the same HLA supertype presenting overlapping peptide repertoires [70,71]. In our study, we can only determine the effect of UVB irradiation on the HLA-C*06:02-restricted autoimmune response against melanocytes. 

Not all patients respond equally to UVB phototherapy [72]. Although HLA association studies on the efficacy of UVB phototherapy are missing, there is indirect evidence that HLA-C*06:02-positive patients may respond better to conventional psoriasis treatment, as the fraction of HLA-C*06:02-positive patients with moderate to severe psoriasis who have failed conventional psoriasis therapies, including phototherapy, is much lower [73]. HLA-B*27^+^ patients not responding to UVB phototherapy would be consistent with the observation that it is the HLA-C-restricted autoimmune response in psoriasis that responds so well to treatment with UVB irradiation [74].

Another limitation is that our experiments have only investigated the short-term molecular changes in the antigen presentation pathway induced by a single UVB irradiation. The incipient recovery of HLA-C expression 24 h after UVB irradiation suggests that the effect of UVB irradiation is temporary. The need for the repeated irradiation of patients to achieve a remission of psoriasis would suggest that the UVB effect is transient, at least initially, and needs to be maintained by continued UVB exposure.

In summary, the present results show that UVB irradiation suppresses the immunogenicity of melanocytes as target cells of the HLA-C*06:02-restricted psoriatic autoimmune response by impairing HLA-C expression and HLA-C-restricted antigen presentation at translational and posttranscriptional levels. Consistently, in a preliminary analysis, the healing of psoriasis through UVB phototherapy was associated with a loss of HLA-C expression on lesional melanocytes. Due to the particularly pronounced effect on HLA-C expression, psoriasis, as an HLA-C-associated T-cell-mediated autoimmune disease, may be particularly susceptible to UVB phototherapy. According to the mechanism of action observed in psoriasis, the suppression of inflammatory IFN-γ-induced HLA-class I expression and antigen presentation may constitute an overall mode of action of UVB therapy that distinguishes T-cell-mediated skin diseases from other inflammatory skin diseases in terms of their susceptibility to the therapeutic efficacy of UVB. These results now form a basis for further targeted investigations to verify that suppressing HLA-class I expression and antigen presentation is a mechanism of action for UVB phototherapy in patients.

## 4. Materials and Methods

### 4.1. Patients

This study was performed in accordance with the Declaration of Helsinki and was approved by the Ethics Committee of the Ludwig-Maximilian-University, Munich (Ref. 151-16). Normal skin specimens for the preparation of primary human melanocytes were derived from the discarded healthy skin of patients undergoing plastic surgery [34]. Biopsies were obtained from untreated chronic psoriasis plaques or a healed psoriasis lesion treated with 41 irradiations with narrow-band UVB with a cumulative dose of 44.1 J/cm^2^ (TL01/120W, UV Therapy System 7002, Waldmann Co., Villingen-Schwenningen, Germany). Informed consent was obtained from all subjects involved in the study.

### 4.2. Cells and Cell Lines

Primary human melanocytes were isolated from the epidermis and expanded as previously described [34,75]. Epidermal single-cell suspensions were expanded in melanocyte growth medium M2 (PromoCell GmbH, Heidelberg, Germany).

The HLA-C*06:02 positive human melanoma cell lines 1205Lu, WM278, and WM793 were originally obtained from the Wistar Institute (Philadelphia, PA, USA). They were cultured in 2% TU medium containing MCDB153, 20% Leibovitz’s L15, 5 µg/mL insulin, 2% FCS, and 1.68 mM CaCl_2_ [34,38]. Generation and cultivation of the Vα3S1/Vβ13S1-TCR hybridoma has been described previously [34,43].

### 4.3. UVB and UVA Irradiation of PHMs or Melanoma Cell Lines

PHMs or melanoma cell lines were seeded in 48-well plates in TU 2% medium at densities of 2.5 × 10^4^ or 5 × 10^4^ cells/well, respectively, and grown to 75–85% confluence. IFN-γ (rHulFN-gamma, Biomol GmbH, Hamburg, Germany) was added to a final concentration of 100 ng/mL for 24 h before UVB or UVA irradiation and addition of Vα3S1/Vβ13S1 TCR hybridoma cells for stimulation experiments. Cells deprived of culture medium were irradiated with 25, 50, 100, or 200 mJ/cm^2^ UVB or 3 or 7.5 mJ/cm^2^ UVA at a distance of 30 cm. For UVB irradiation, we used a Waldmann UV-800 unit (Waldmann Co., Villingen-Schwenningen, Germany) fitted with Philips TL-20W/12 fluorescent lamps emitting UVB with a peak emission at 310–315 nm. A Dr. Hönle dermalight UltraVA1 unit 2 kW/4 kW (Dr. Hönle Co., Gilching, Germany) having an emission spectrum of 340–440 nm was used for UVA irradiation. The irradiation dose was measured with a UV radiometer (Eisai Co., Tokyo, Japan).

### 4.4. Cell Viability and Apoptosis Assay

Each 2 × 10^5^ cells/well were seeded in 6-well plates and irradiated with UVB 20, 60, and 200 mJ/cm^2^. After 24 h, cell viability was assessed using the Trypan blue assay under an inverted microscope. To determine apoptosis, cells were detached 24 h after UVB exposure using EDTA/PBS, pelleted by centrifugation (300 g, for 5 min) and re-suspended in annexin V binding buffer (10 mM HEPES, 140 mMNaCl, 2.5 mM CaCl2), containing 1 μg/mL Cy™ 5 annexin V binding (BD Biosciences, Oxford, UK) and 10 μg/mL propidium iodide (Sigma-Aldrich, St. Louis, MO, USA) for 15 min at room temperature in the dark. Cells were analyzed by flow cytometry (BD Biosciences FACSCanto, FACSDiva software v5.0.3), and data were analyzed using FlowJo software version 10.8.1 (Treestar, Ashland, OR, USA).

### 4.5. Vα3S1/Vβ13S1-TCR Hybridoma Activation Assays and Flow Cytometry Analysis

Immediately after UV irradiation, hybridoma cells (1 × 10^5^ cells per well) were added. sGFP induction in Vα3S1/Vβ13S1-TCR hybridoma cells was examined after 24 h of co-culture with PHMs or melanoma cell lines by UV fluorescence microscopy and/or flow cytometry [34]. As negative/positive controls, Vα3S1/Vβ13S1-TCR hybridoma cells were incubated in culture plates either untreated or pre-coated with CD3 antibody (17A2, 2 µg/mL in PBS; eBioscience, San Diego, CA, USA,) [34]. Cells were analyzed on a BD FACS Calibur Flow Cytometer (BD Bioscience, San Jose, CA, USA). Data were analyzed by Flow Jo software v10.8.1 (Tree Star, Ashland, OR, USA).

### 4.6. Evaluation of HLA Expression

Detached PHMs or melanoma cell lines were stained either with HLA-C antibody (DT-9, BD Bioscience #566372), HLA-ABC antibody (W6/32, #311406, BioLegend, San Diego, CA, USA), or corresponding isotype IgG (IgG2b, IgG2a BioLegend #401208, #400214), all conjugated with phycoerythrin. Data were acquired by flow cytometry and analyzed by FlowJo software [38].

Primary human melanocytes on chamber glass slides coated with 0.5 mg/mL poly-D-lysine (#P7280, Sigma Aldrich) were stained with antibodies for HLA-C, HLA-ABC, or corresponding isotype control (BioLegend #401216, #400264, #311441) and anti-mouse IgG (H + L) Alexa Fluor 488 or 594 secondary antibody (Invitrogen #A11001, #A11005) [34].

### 4.7. RNA Isolation, Reverse Transcription, and Real-Time PCR

Total RNA was extracted using RNeasy Universal kit (Qiagen GmbH, Hilden, Germany) and reverse transcribed into cDNA using SuperScript^®^ III Reverse Transcriptase (Invitrogen by Thermo Fisher Scientific, Schwerte, Germany) according to the manufacturer’s instructions.

For ADAMTSL5, B2M and the house keeping gene PBGD, and for STAT1, NLRC5, IRF1, Oct1, ERAP1, HLA-C (gene expression assay Hs03044135_m1), and the house keeping gene GADPH, TaqMan qPCR was carried out using LightCycler^®^ 2.0 System (Roche, Grenzach-Whylen, Germany) with the TaqMan universal PCR master mix (Roche). For miR-148a and its housekeeping gene RNU6B, predesigned TaqMan™ Advanced miRNA assays (hsa-miR-148a, catalog #4427975, assay ID 000470; RNU6B, catalog #4427975, assay ID 001093) and the TaqMan Micro RNA reverse Transcription Kit (catalog #4366596, all from Thermo Fisher Scientific, Schwerte, Germany) were employed. The relative expression level of each mRNA transcript was determined using the comparative method (ΔΔCt method) [76,77]. Quantitative PCR was performed in triplicates. The light cycler primers are given in Table 1.

### 4.8. siRNA Knockdown Experiments

Predesigned siRNAs for NLRC5, HLA-C and B2M were obtained from Ambion (now Thermo Fisher Scientific). siRNA transfection of melanoma cell lines was performed using Invivofectamine (Invitrogen). 24 h later, HLA-C expression and hybridoma stimulation were analyzed. The sequences of the oligonucleotides are given in Table 2.

### 4.9. Immunofluorescence Staining of Paraffin Sections

Following heat-induced antigen retrieval with Tris-EDTA buffer (10 mM Tris Base, 1 mM EDTA, and 0.05% Tween 20, pH 9.0) at 120 °C for 15 min, sections were incubated with primary antibody for HLA-C and c-kit or isotype controls at 4 °C for 72–96 h, washed, and incubated with fluorescence-conjugated secondary antibody for 90 min. Nuclei were counterstained with DAPI. To stain adherenT-cells, chamber glass slides were coated with 0.5 mg/mL poly-D-lysine (# P7280; Sigma-Aldrich) at 4 °C overnight and seeded with primary human melanocytes adjusted to yield comparable cell density on analysis. After 2 d of culture, cells were reacted with Abs for HLA-C, HLA-ABC, or corresponding isotype control (no. 401216, no. 400264, no. 311441, and DT-9, all without azide; BioLegend) and anti-mouse IgG (H1L) Alexa Fluor 488 or 594 secondary Ab (no. A11001 or no. A11005; Invitrogen). After washing, the cells were mounted with fluorescence mounting medium (no. S3203; DAKO), as described in [34,38].

Microscopic analysis was performed on an Axio Observer microscope (Carl Zeiss, Oberkochen, Germany) with a Chroma ET filter-set and a Photometrics CoolSNAP-HQ2 digital camera system (Cambridge Scientific, Watertown, MA, USA). Images were merged using MetaMorph Imaging Software v7.7.6 (Molecular Devices, LLC., San Jose, CA, USA). Epidermal areas of skin sections were completely photo documented at a 100-fold magnification with GFP, Texas Red, and DAPI filters, respectively (Visitron Systems GmbH, Puchheim, Germany), and images were merged. Epidermal segments were evaluated in each photographic image using (National Institutes of Health, Bethesda, MD, USA). Stained cells were counted in each overlay. Resulting median values in each individual were used for statistical analyses. All antibodies employed in the study are given in Table 3.

### 4.10. Statistical Analysis

For statistical analysis of melanoma cell lines, the two-tailed student’s *t*-test was used. The data were analyzed using XLSTAT Software version 2014.05.03 (Addinsoft, Paris, France). A *p*-value less than 0.05 was considered statistically significant, * *p* < 0.05, ** *p* < 0.01 and *** *p* < 0.001.

For statistical analysis of human specimens, a two-group comparison was performed using the Mann–Whitney U test. *p* < 0.05 was considered significant. All statistical analyses were performed using XLSTAT Software version 2014.05.03 (Addinsoft). Sample size was determined based on preliminary data (mean and variation) and previous publications, as well as observed effect sizes. No samples were excluded from analysis.

## 5. Conclusions

Suppressing the expression of HLA-class I molecules is a mechanism of action that may render T-cell-mediated inflammatory skin diseases sensitive to UVB phototherapy. Our findings further suggest targeting the expression of HLA-C*06:02 as a strategy for the development of new therapeutic concepts for psoriasis.

## Figures and Tables

**Figure 1 ijms-26-02858-f001:**
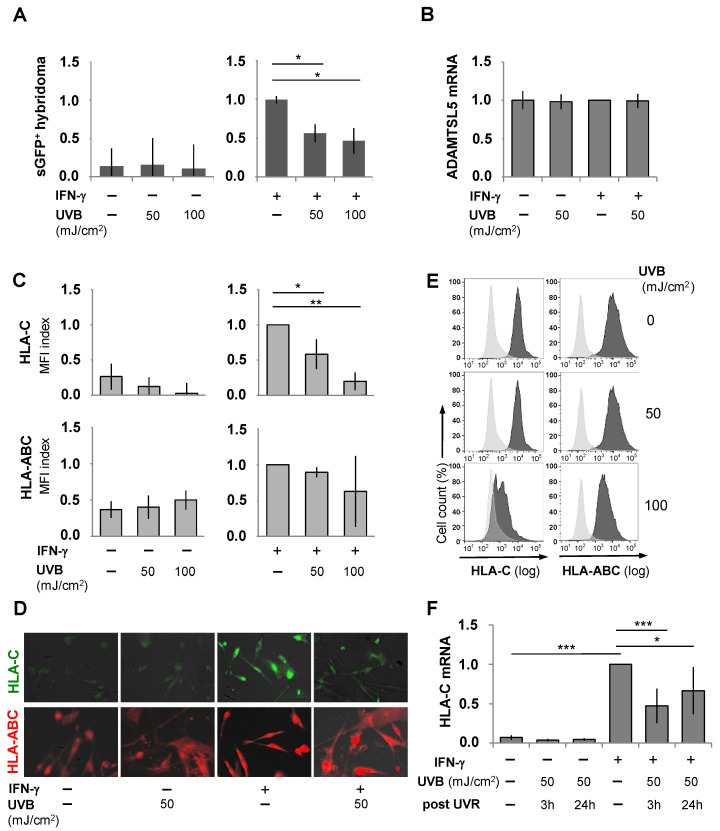
UVB irradiation reduces the immunogenicity of IFN-γ-conditioned melanocytes for the autoreactive melanocyte-specific Vα3S1/Vβ13S1 TCR. (**A**) Stimulation of the Vα3S1/Vβ13S1 TCR hybridoma cells by HLA-C*06:02^+^ PHMs grown in the presence or absence of IFN-γ and effect of irradiation with 50 or 100 mJ/cm^2^ UVB. Induction of sGFP was determined after 24 hrs of co-culture by flow cytometry. (**B**) Effect of IFN-γ and UVB irradiation on relative ADAMTSL5 transcript levels as assessed by qPCR. Water served as a negative control. (**C**) Effect of UVB-irradiation on spontaneous or IFN-γ-induced expression of HLA-C or HLA-ABC determined by flow cytometry or (**D**) fluorescence microscopy. Mean fluorescence intensity (MFI) of HLA-C (DT9) or HLA-ABC (W6/32) in (**C**) is normalized to the IFN-γ-induced MFI of each analysis. (**E**) Representative flow cytometry analyses of the effect of UVB irradiation on the IFN-γ-induced expression of HLA-C or HLA-ABC with isotype controls (light gray). (**F**) Effect of IFN-γ and UVB irradiation on relative HLA-C transcript levels in PHMs assessed by qPCR. Water served as negative control. In all experiments, data are normalized to IFN-γ-exposed PHMs set as 1. Data summarizing technical triplicates from three or more independent experiments are shown as mean  ±  s.e.m. and compared by Mann–Whitney U-test (* *p* < 0.05, ** *p* < 0.01, *** *p* < 0.005).

**Figure 2 ijms-26-02858-f002:**
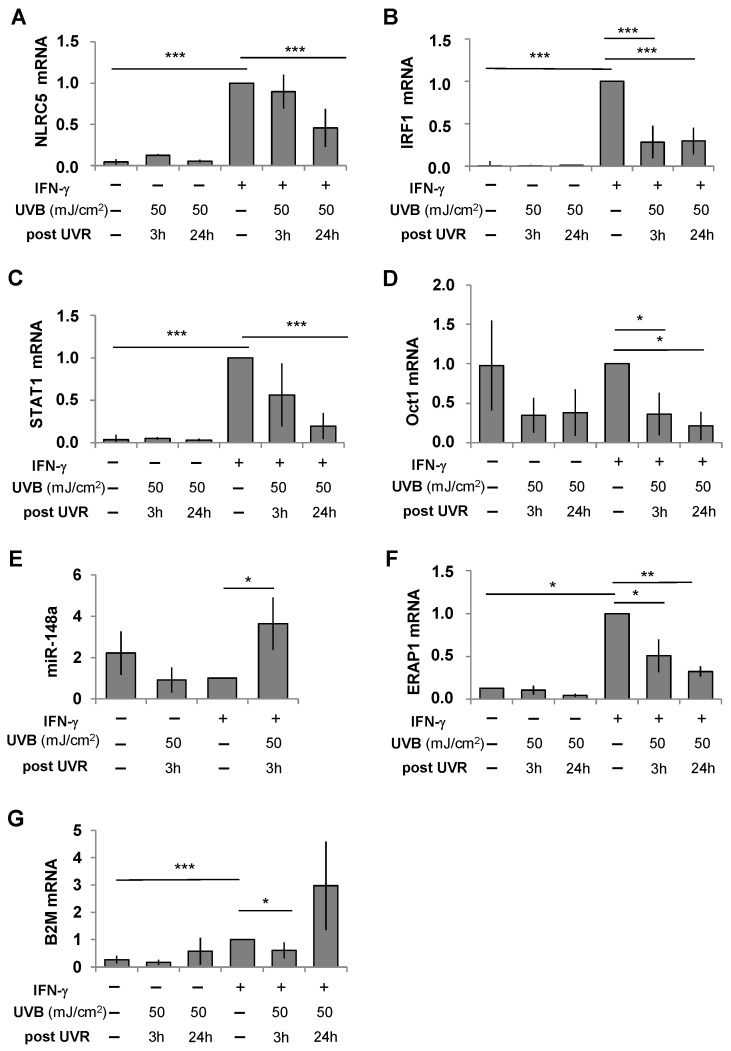
UVB affects key transcriptional and posttranscriptional regulators of HLA-class I and HLA-C expression in human melanocytes. Effect of UVB irradiation on the relative transcript levels of (**A**) NLRC5, (**B**) IRF1, (**C**) STAT1, (**D**) Oct1, (**E**) miR-148a, (**F**) ERAP1, and (**G**) B2M in PHMs as determined by qPCR three or 24 hrs after irradiation with UVB, 50 mJ/cm^2^. Results are normalized to IFN-γ-induced gene transcription without UVB irradiation set as 1. Data summarizing technical triplicates from each three independent experiments are shown as mean  ±  s.e.m. and compared by Mann–Whitney U-test (* *p* < 0.05, ** *p* < 0.01, *** *p* < 0.005).

**Figure 3 ijms-26-02858-f003:**
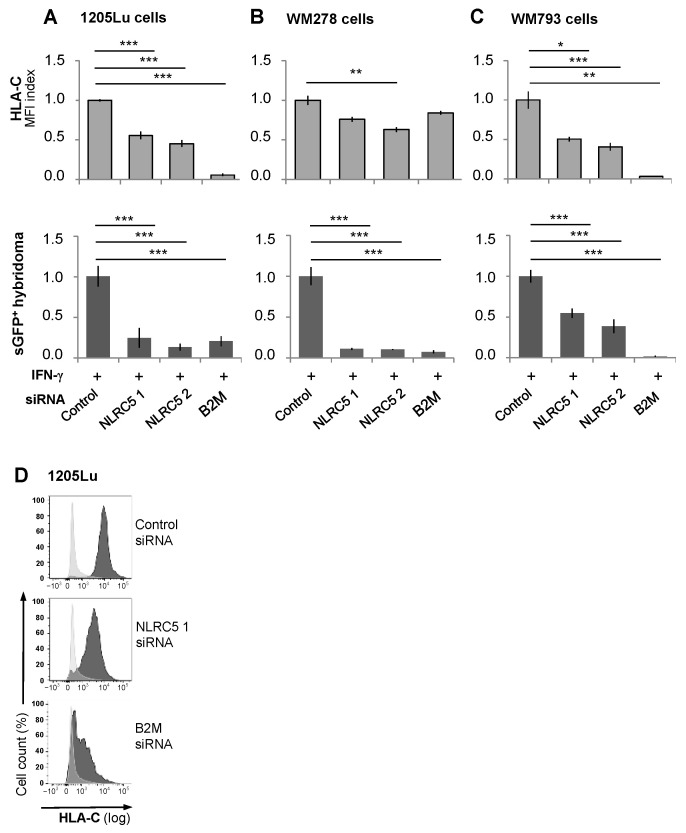
Knockdown of NLRC5 or B2M in melanoma cell lines downregulates HLA-C expression and reduces Vα3S1/Vβ13S1 TCR stimulation. Effect of siRNA knockdown of NLRC5 or B2M in 1205Lu (**A**), WM278 (**B**), or WM793 cells (**C**) on the expression of HLA-C (upper graphs) and the stimulation of the Vα3S1/Vβ13S1-TCR hybridoma (lower graphs). 24 h after siRNA transfection, cells were incubated with IFN-γ. HLA-C expression was analyzed 24 hrs later by DT-9 antibody staining and flow cytometry. For TCR hybridoma stimulation, Vα3S1/Vβ13S1-TCR hybridoma cells were added, and the induction of sGFP measured after 24 h of co-culture by flow cytometry. Data are normalized to the results obtained with control siRNA set as 1. Representative flow cytometry analyses show the effect of knockdown of NLRC5 or B2M (**D**) on HLA-C expression, with isotype control in light gray. Data summarizing technical triplicates are shown as mean  ±  s.e.m. and compared by Mann–Whitney U-test (* *p* < 0.05, ** *p* < 0.01, *** *p* < 0.005).

**Figure 4 ijms-26-02858-f004:**
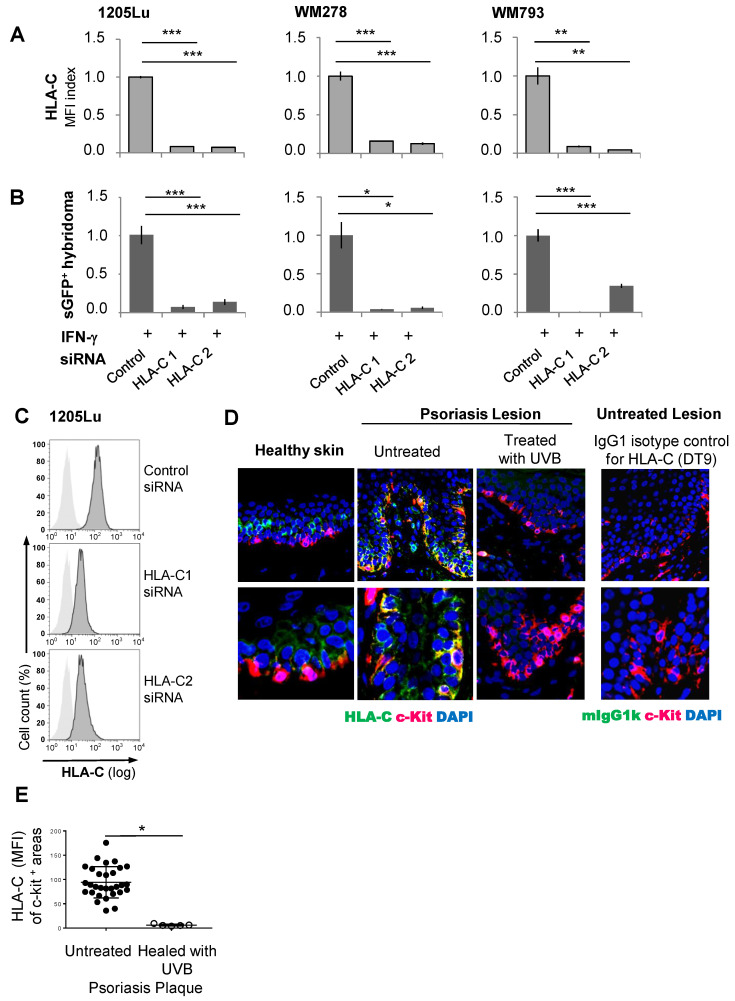
Impaired Vα3S1/Vβ13S1-TCR stimulation by knockdown of HLA-C and reduced HLA-C expression on melanocytes following UVB phototherapy of psoriasis. Effect of siRNA knockdown of HLA-C in 1205Lu, WM278 and WM793 cells on HLA-C expression (**A**) and Vα3S1/Vβ13S1 TCR hybridoma stimulation (**B**). Data summarizing technical triplicates of three experiments are shown as mean  ±  s.e.m. and compared by Mann–Whitney U-test (* *p* < 0.05, ** *p* < 0.01, *** *p* < 0.005). The experimental procedure is the same as in Figure 3. Data are normalized to IFN-γ-exposed cell lines set as 1. (**C**) Representative flow cytometry analyses of HLA-C expression following siRNA-knockdown. (**D**) Expression of HLA-C (DT9, green) on melanocytes (c-Kit, red) in biopsy sections from healthy skin and a psoriasis plaque before and after UVB-phototherapy (cumulative dose of 44.1 J/cm^2^) in 100- and 200-fold magnification. The overlay of green and red in merged images is given in yellow. Mouse IgG1k served as isotype control. (**E**) Mean fluorescence intensity (MFI) of HLA-C in c-kit-positive areas of untreated (*n* = 5) or healed (*n* = 1) psoriasis plaques. Each dot represents measurement of one visual field at a 100-fold magnification. Statistical analysis was performed using the Mann–Whitney U test (* *p* < 0.05).

**Table 1 ijms-26-02858-t001:** Light cycler primers (qPCR).

**Gene**	**Forward**	**Reverse**
IRF 1	5′ ggc aca tcc cag tgg aag 3′	5′ ccc ttc ctc atc ctc atc tgt 3′
Oct1	5′ tcc tct tcc tgc tct act act gg	5′ tgg tcc att atc ttt att gct tca 3′
B2M	5′ ttc tgg cct gga ggc tat c 3′	5′ tca gga aat ttg act ttc cat tc 3′
ADAMTSL5	5′ tac cag tgg gtg ccc ttc 3′	5′ ggc cga agc tgt ggt aga 3′
ERAP 1	5′ gtc act gtg aag atg agc acc ta 3′	5′ tgt ctg gca cag cat aaa cag 3′
NLRC5	5′ ggt gct gct gag tac ttt 3′	5′ ggt tgg ctt ttc ccc tca 3′
STAT 1	5′ gag ctt cac tcc ct tagt ttt ga 3′	5′ cac aac ggg cag aga ggt 3′
**Gene**	**Product details**	**Locator**
miR-148a	Assay ID Hs04273238_s1	See Thermo Fisher Scientific—TaqMan Search (last accessed 22 February 2025)
HLA-C	Assay ID Hs03044135_m1	See Thermo Fisher Scientific—TaqMan Search (last accessed 22 February 2025)

**Table 2 ijms-26-02858-t002:** siRNA oligonucleotides.

siRNA	Sense Strand	Antisense Strand	siRNA Details
NRLC5-1	GCUGAUCUUUGAUGGGCUAtt	UAGCCCAUCAAAGAUCAGCag	Chr.16: 56989547-57083524 on Build GRCh38
NRLC5-2	GAGCUGGACUUGUCUAACAtt	UGUUAGACAAGUCCAGCUCct	Chr.16: 56989547-57083524 on Build GRCh38
HLA-C-1	GUCAAUUCCUAGAAGUUGAtt	UCAACUUCUAGGAAUUGACtt	Chr.6: 31268749-31272136 on Build GRCh38
HLA-C-2	GUUGAGAGAGCAAAUAAAGtt	CUUUAUUUGCUCUCUCAACtt	Chr.6: 31268749-31272136 on Build GRCh38
B2M	CAUCCGACAUUGAAGUUGAtt	UCAACUUCAAUGUCGGAUGga	Chr.15: 44711487-44718159 on Build GRCh38

**Table 3 ijms-26-02858-t003:** Antibodies.

Antibody		Origin	Isotype	Designation	Distributor	Order Number
HLA-C	PE	mouse	IgG2b, k	DT-9	Bio Legend, Inc	AB-2650941
HLA-ABC	PE	mouse	IgG2a, k	W6/32	Bio Legend, Inc	311406
c-kit		Rabbit			Dako	A4502
Anti-mouse IgG	Alexa 488	Goat			Invitrogen	A11001
Anti-rabbit IgG	Alexa 594	Goat			Invitrogen	A11037
Mouse IgG2b, k Isotype	PE	mouse	IgG2b, k	MG2b-57	Bio Legend, Inc	401208
Mouse IgGa, k Isotype	PE	mouse	IgG2a, k	eBM2a	eBiosience	12-4724-42
Mouse IgG1, k Isotype		mouse	IgG1, k	MG1-45	Bio Legend, Inc	401402
Human BD Fc Block					BD Bioscience	564219

## Data Availability

All data are available in the main text or the Appendix A.

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
