# Peer review of "Down-Regulation of HLA-C Expression on Melanocytes May Contribute to the Therapeutic Efficacy of UVB Phototherapy in Psoriasis"

_ijms, 2025, doi:10.3390/ijms26072858_

Round 1
Reviewer 1 Report
Comments and Suggestions for Authors
The authors propose that UVB irradiation reduces HLA-C expression on melanocytes by downregulating key transcriptional regulators, confirmed in vivo through psoriasis patient biopsies. HLA-C was highly expressed in lesional melanocytes but significantly reduced after UVB therapy, suggesting a possible mechanism for its clinical effects. The study supports the hypothesis that HLA-C06:02 mediates an autoimmune response in psoriasis by presenting a melanocyte autoantigen to CD8+ T cells, providing insights into UVB phototherapy’s role in psoriasis treatment.
- While intriguing, the involvement of non-cytotoxic CD8+ T cells in psoriasis requires further validation, especially given the presence of post-inflammatory hyperpigmentation.
- Not all psoriasis patients have HLA-C*06:02, yet UVB therapy remains broadly effective, suggesting alternative or additional mechanisms beyond melanocyte antigen presentation.
- It remains unclear whether reducing HLA-C directly decreases CD8+ T cell activation in psoriasis lesions, necessitating further functional studies
- While in vivo biopsy data provide initial evidence of UVB-induced HLA-C suppression, additional studies are required to confirm its direct effect on immune modulation.
- The title suggests a direct connection between UVB-induced HLA-C downregulation and psoriasis improvement. However, the study primarily shows that UVB reduces HLA-C expression in melanocytes, but does not establish this as the primary mechanism for psoriasis resolution. While UVB may influence psoriasis by modulating HLA-C expression, additional mechanisms likely contribute to its therapeutic effects.
Author Response
Plese see the response to the comments of Reviewer 1 in the enclosed file.

Reviewer 2 Report
Comments and Suggestions for Authors
Dear Authors,
This manuscript explores how UVB therapy helps treat psoriasis by suppressing IFN-γ-induced HLA-C expression and antigen presentation. Using an in vitro model, the authors show that UVB reduces melanocyte immunogenicity by inhibiting the STAT1-IRF1-NLRC5 pathway and upregulating miR-148a, which in turn downregulates HLA-C. Suggesting that UVB therapy’s effectiveness in T-cell-mediated skin diseases may stem from its ability to suppress HLA-C-driven antigen presentation.
This study is well-structured, with strong experimental design, solid data analysis, and convincing conclusions. The authors use a range of techniques, including TCR activation assays, qPCR, flow cytometry, and immunohistochemistry, to support their claims. The findings are both novel and clinically relevant, offering new insights into how UVB selectively affects psoriasis, a disease strongly associated with HLA-C*06:02, and potentially identifying new therapeutic targets.
However, I have several suggestions to further improve the manuscript:
1) In Section 2.4 of the Results, the authors use Figure 1F and Supplementary Figures S2D and S3D to demonstrate that IFN-γ significantly increases HLA-C mRNA levels in primary human melanocytes (PHMs) and melanoma cell lines, which then decrease significantly as early as 3 hours after UVB irradiation. However, I noticed that in Figure 1F, at 24 hours post-UVB treatment, HLA-C mRNA levels appear to recover or stabilize. This raises the question of how long this effect persists. Does the suppression last for multiple days? Would increasing the UVB dose affect the recovery of HLA-C expression?
I suggest including a broader time-course analysis (e.g., 6 h, 12 h, 24 h, 48 h) to evaluate the long-term effects of UVB on HLA-C expression and T cell responses. This would help determine whether UVB-mediated suppression of HLA-C is transient or sustained and whether higher UVB doses could influence its recovery dynamics.
2) In Section 2.4 of the Results, the authors suggest that “the UVB-induced reduction in HLA-C expression in the presence of IFN-γ” is due to the increased expression of miR-148a. However, the cited references supporting the claim that "miR-148a can suppress HLA-C allele expression" do not provide direct experimental evidence from cellular studies. Instead, they primarily rely on population genetic analyses and bioinformatics predictions, rather than functional experiments that directly validate miR-148a’s inhibitory effect on HLA-C.
To strengthen this conclusion, I recommend performing miRNA mimic or inhibitor experiments to overexpress or inhibit miR-148a, which would confirm whether miR-148a plays a direct role in UVB-mediated HLA-C downregulation. Otherwise, the observed HLA-C reduction may not be sufficiently explained by miR-148a alone.
3) In Section 2.5 of the Results, the actual immunological effects of UVB therapy in psoriasis patients require further clinical validation. While the study analyzes changes in HLA-C expression in melanocytes before and after UVB treatment, it does not assess T cell activation status post-treatment, such as whether IL-17 and IFN-γ production decreases.
I recommend incorporating additional immunostaining data using patient samples. Specifically, immunofluorescence (IF) or immunohistochemistry (IHC) staining could be used to evaluate the activation status of CD8+ T cells in psoriatic lesions before and after UVB treatment, focusing on markers such as IL-17 and IFN-γ. This would provide stronger evidence for the immunosuppressive effects of UVB on T cell responses.
4) In the manuscript, three melanoma cell lines were used to support the findings in PHMs. However, I noticed that some conclusions were not validated across all three cell lines. For example, Supplementary Figures S2B and S3B were used to demonstrate that IFN-γ and UVB irradiation significantly affect the mRNA transcription levels of the melanocyte autoantigen ADAMTSL5, yet the quantification of ADAMTSL5 expression in the WM278 cell line was not provided. This inconsistency also contributes to a lack of uniformity in the supplementary figures. I recommend including the quantification of ADAMTSL5 expression in the WM278 cell line to ensure completeness. Additionally, Supplementary Figure S4B should be moved to S5, and the missing UVA treatment results for the other two cell lines should be added. Similarly, the quantification of HLA-C expression in the WM278 cell line should be included. These changes would improve the consistency and clarity of Supplementary Figures S2, S3, and S4, ensuring a uniform format across all datasets.
5) In Section 2.3 of the Results, the authors analyzed the expression of HLA-class I molecules (HLA-ABC) and HLA-C using flow cytometry and fluorescence microscopy. They stated that “IFN-γ treatment induces upregulation of HLA-class I molecules in PHMs (Figure 1C) and melanoma cell lines (Supplementary Figures S2C, S3C, S4C), with HLA-C being particularly prominent.” However, I noticed a discrepancy in the HLA-ABC results between PHMs and the three melanoma cell lines. In PHMs, HLA-ABC expression does not show significant changes, whereas in all three melanoma cell lines, HLA-ABC levels decrease to varying degrees. This observation suggests that UVB might regulate HLA-ABC differently in normal melanocytes and transformed melanoma cells.
I recommend adding a discussion on this phenomenon in the Discussion section to explore possible reasons for this difference. Potential factors could include differences in baseline HLA-ABC expression, epigenetic regulation, antigen processing machinery, or cell-specific sensitivity to UVB-induced immune modulation.
Addressing the above points would further strengthen the manuscript by enhancing the mechanistic understanding of UVB’s effects on the STAT1-IRF1-NLRC5 pathway and its clinical implications.
Best,
Author Response
Please see my response to the comments of Rveiwer 2 in the enclosed file.

Round 2
Reviewer 2 Report
Comments and Suggestions for Authors
Dear Author,
I have carefully reviewed the revised version of your manuscript. I am happy to see that you have made several changes based on my previous suggestions and have clarified certain issues. I believe your manuscript has now reached the required standard for publication.
Best,